# VISUAL CoT MAKES VLMS SMARTER BUT MORE FRAGILE

## ABSTRACT

Chain-of-Thought (CoT) techniques have significantly enhanced reasoning in Vision-Language Models (VLMs). Extending this paradigm, Visual CoT integrates explicit visual edits, such as cropping or annotating regions of interest, into the reasoning process, achieving superior multimodal performance. However, the robustness of Visual CoT-based VLMs against image-level noise remains unexplored. In this paper, we present the first systematic evaluation of Visual CoT robustness under visual perturbations. Our benchmark spans 12 image corruption types across 4 Visual Question Answering (VQA) datasets, enabling a comprehensive comparison between VLMs that use Visual CoT, and VLMs that do not. The results reveal that integrating Visual CoT consistently improves absolute accuracy regardless of whether the input images are clean or corrupted by noise; however, it also increases sensitivity to input perturbations, resulting in sharper performance degradation compared to standard VLMs. Through extensive analysis, we identify the intermediate reasoning components of Visual CoT, i.e., the edited image patches , as the primary source of fragility. Building on this analysis, we propose a plug-and-play robustness enhancement method that integrates Grounding DINO model into the Visual CoT pipeline, providing high-confidence local visual cues to stabilize reasoning. Our work reveals clear fragility patterns in Visual CoT and offers an effective, architecture-agnostic solution for enhancing visual robustness.

## 1 INTRODUCTION

With the introduction of Chain-of-Thought (CoT) techniques, Large Language Models (LLMs) have achieved remarkable progress in reasoning capabilities. Recent studies have extended CoT to Vision-Language Models (VLMs), evolving from CoT pipelines that rely solely on textual reasoning to Visual Chain-of-Thought (Visual CoT) approaches that incorporate visual information into the reasoning process (Shao et al., 2024; Jiang et al., 2025a; Wang et al., 2025; Chen et al., 2024; Fu et al., 2025), thereby significantly enhancing multimodal reasoning performance. For Visual CoT methods, a common practice is to perform visual editing on the input images, such as cropping, annotating, or modifying regions of interest. The edited and original images are jointly fed into the model, which is guided to perform step-by-step reasoning based on both visual inputs, thus supporting finer-grained multimodal understanding.

However, despite several studies exploring the robustness of CoT-based VLMs under purely textual reasoning scenarios (Jiang et al., 2025a; Zhou et al., 2024a; Jiang et al., 2025b; Wang et al., 2024), there has been no systematic investigation into the robustness of Visual CoT-based VLMs. Unlike standard textual CoT, Visual CoT introduces explicit visual manipulations, which inherently interact with the reasoning process. Under noisy conditions, these added components may amplify the effects of input perturbations, making the system more susceptible to errors and raising new challenges for multimodal reasoning (as shown in Figure 1).

To address this gap, we propose a robustness evaluation framework for Visual CoT-based VLMs, aiming to systematically assess how Visual CoT affects model robustness under visual perturbations. Specifically, we employ 12 distinct visual perturbation techniques, systematically applied to the input images. To quantify the robustness impact introduced by Visual CoT, we compute the performance degradation separately under two paradigms, Visual CoT-enhanced VLMs and standard VLMs without CoT, by comparing outputs before and after perturbation. This setup enables a

Figure 1: Visual CoT pipeline amplifies the effects of input noise due to intermediate visual editing steps, where noise influences both global and local components, in contrast to Standard VLMs where noise only affects a single input stage.

direct and systematic analysis of how incorporating Visual CoT changes robustness in multimodal reasoning tasks.

Our experimental results show that while Visual CoT VLMs consistently achieve higher absolute accuracy across all perturbation conditions, their performance degrades more sharply under noisy inputs compared to standard VLMs. Further experiments reveal that noise compromises the reliability of intermediate reasoning components, which subsequently propagates to the final stage and leads to more severe accuracy degradation. Moreover, attention analysis demonstrates that Visual CoT VLMs exhibit a more concentrated attention distribution compared to standard VLMs. These attention characteristics explain why Visual CoT VLMs achieve higher accuracy under whether clean or perturbed inputs.

Building on this analysis, we propose a lightweight, plug-and-play robustness enhancement strategy by integrating Grounding DINO model (Liu et al., 2023b) into the Visual CoT pipeline. Grounding DINO automatically identifies high-confidence image regions relevant to the question and injects these localized cues into the reasoning chain. This provides the model with richer and redundant visual information, effectively mitigating the adverse effects of perturbation without requiring additional fine-tuning or architecture modification.

This paper makes several contributions to the literature: ❶We present the first comprehensive evaluation of Visual CoT-based VLMs under visual perturbations, offering new insights into the trade-off between their accuracy and robustness. ❷We introduce a lightweight strategy that integrates Grounding DINO into the Visual CoT pipeline, substantially improving reasoning stability under noisy conditions. ❸We establish a robustness evaluation benchmark that spans 12 visual perturbation types across multiple datasets on VQA task, providing a reproducible and extensible foundation for future research on Visual CoT robustness.

## 2 RELATED WORK

### 2.1 CHAIN-OF-THOUGHT IN VISION-LANGUAGE MODELS

CoT prompting has significantly advanced the reasoning capabilities of LLMs by decomposing complex problems into intermediate steps. Building on this success, recent research has extended CoT to VLMs, enabling multimodal reasoning by integrating visual evidence into the reasoning chain. Zhang et al. (2024) first formally introduce the concept of Multimodal-CoT (MCoT) and extend it into a rationalizing-answering stages paradigm. Yang et al. (2023) introduce MM-REACT, which combines LLMs with vision experts through prompt-based coordination, enabling zero-shot multimodal reasoning across diverse visual tasks.

Later, the development of CoT gradually evolved from purely textual prompting to frameworks that integrate explicit visual editing into the reasoning process. Shao et al. (2024) introduce a large-scale visual CoT dataset with bounding boxes and reasoning steps, enabling VLMs to identify key regions

and enhance multimodal reasoning. Hu et al. (2024) propose Sketchpad, enabling VLMs to sketch visual artifacts during reasoning and improving performance on complex visual and mathematical tasks. Fu et al. (2025) introduce ReFous, which equips VLMs with visual editing capabilities to generate "visual thoughts", achieving substantial gains in structured image understanding. These studies highlight the progression of CoT in VLMs, from textual prompting to Visual CoT pipelines incorporating editing and sketching, broadening the scope and effectiveness of multimodal reasoning.

## 2.2 ROBUSTNESS OF CHAIN-OF-THOUGHT

While CoT has achieved remarkable performance gains, its robustness under input perturbations remains an open challenge. Zhou et al. (2024b) address the challenge of noisy rationales in CoT prompting by introducing the NoRa dataset and proposing CD-CoT, a contrastive denoising method that significantly improves reasoning robustness under irrelevant or inaccurate intermediate thoughts. Jin et al. (2024) investigate the security vulnerabilities of CoT-based models in code generation and propose SABER, a model-agnostic backdoor attack leveraging self-attention, demonstrating that CoT models remain highly susceptible to stealthy data poisoning. Wang et al. (2024) reveal that CoT-based MLLMs exhibit only limited resistance to adversarial attacks despite their multi-step reasoning process.

Collectively, these studies reveal that despite the strong reasoning capabilities of CoT-based systems, they remain vulnerable to various forms of security threats. However, prior studies have predominantly investigated robustness in purely textual CoT prompting frameworks, the robustness of Visual CoT approaches remains largely underexplored. In this work, we aim to bridge this gap by conducting a comprehensive study on the robustness of Visual CoT reasoning under diverse noisy and adversarial conditions.

## 3 METHODOLOGY

To systematically examine the impact of incorporating Visual CoT on model robustness, we compare two paradigms: (1) Standard VLMs, which generate answers directly from image–question pairs, and (2) Visual CoT VLMs, which explicitly integrate intermediate visual reasoning steps. Our evaluation focuses on how their performance degrades when subjected to identical perturbations applied to the input images.

### 3.1 TWO VLM PARADIGMS

In this study, we compare two modeling paradigms:

**1) Standard VLMs** The Standard VLMs refers to the conventional multimodal question answering framework, where the model receives an image–question pair and directly generates the final answer. The reasoning process is entirely implicit within the model's internal representations, without interpretable intermediate steps.

**2) Visual CoT VLMs** In contrast, Visual CoT VLMs introduce intermediate reasoning stages into the multimodal pipeline. We adopt VisCoT (Shao et al., 2024) as a representative implementation. Given an image–question pair, VisCoT first predicts a bounding box for the most relevant region, and then crops the corresponding patch. This patch and the original image are jointly encoded, and their visual features are fused with the textual input to produce the final answer.

### 3.2 PERTURBATION DESIGN

To evaluate the robustness of Standard and Visual CoT VLMs under noisy conditions, we design controlled perturbation experiments by applying both natural corruptions and adversarial attacks to the visual inputs and measuring model performance degradation across different perturbation types.

For natural perturbations, we adopt image-level corruption strategies from the ImageNet-C benchmark (Hendrycks & Dietterich, 2019), which cover a broad range of common distortions. We select 8 types grouped into four categories: (1) Noise: Gaussian Noise, Shot Noise, Impulse Noise; (2)

Blur: Defocus Blur, Zoom Blur; (3) Digital: Pixelate, Elastic Transformations, Contrast Adjustments. Each corruption is applied at 5 severity levels, enabling fine-grained analysis of performance degradation under increasing noise intensity. The detailed description about severity setting is shown in Appendix A.2.

In addition, we incorporate widely used white-box adversarial attacks, including FGSM (Goodfellow et al., 2014), BIM (Kurakin et al., 2017) , PGD (Makelov et al., 2025) and C&W (Carlini & Wagner, 2016). In our white-box attack setup, the generation of adversarial examples for Visual CoT VLMs follows the same strategy as for Standard VLMs. Specifically, we only apply adversarial perturbations to the initial input image, without modifying intermediate localized patches. This ensures a fair comparison between the two paradigms by keeping the attack surface and perturbation budget consistent. We also manually define 5 severity levels for each attack based on parameters (e.g., iteration count, step size) that determine the strength and impact of the perturbation. The detailed implementation procedures of these algorithms, along with the generated adversarial examples, are provided in the Appendix A.3.

## 3.3 Robustness Evaluation Metrics

We adopt Visual Question Answering (VQA) as the primary evaluation task, as it is a representative and widely used benchmark for assessing multimodal reasoning capabilities. Formally, the VQA task is defined as follows:

Given a natural language question $q$ and an associated image $i$ as the visual context, the model is required to generate an answer. Each question is paired with a ground truth answer $gt$, which serves as the reference for evaluation. Our evaluation dataset $\mathcal{D}_{\text{eval}}$ consists of triplets $(q, i, gt)$. For a given Vision-Language Model $f$ that takes $(q, i)$ as input and outputs an answer $f(q, i)$, we define the **Answer Accuracy** over $\mathcal{D}_{\text{eval}}$ as:

$$\text{Acc}(f, \mathcal{D}_{\text{eval}}) \stackrel{def}{=} \frac{1}{|\mathcal{D}_{\text{eval}}|} \sum_{(q,i,gt) \in \mathcal{D}_{\text{eval}}} \mathbb{1}(f(q, i), gt),$$

where $\mathbb{1}(\cdot)$ is the indicator function, returning 1 if $f(q, i)$ exactly matches $gt$, and 0 otherwise. In our evaluation, the indicator function is implemented by GPT-4O acting as an automatic evaluator, which compares the predicted and ground truth answers while accounting for minor paraphrasing or synonym variations.

To evaluate model robustness, we apply input-level perturbations during inference by directly corrupting the image. Formally, given a perturbation operator $\delta(\cdot)$, the perturbed evaluation set is defined as $\delta(\mathcal{D}_{\text{eval}}) = \{(\delta(q_k), i_k, gt_k)\}_{k=1}^{N}$, where $\delta$ is applied to the image in each sample of the original dataset $\mathcal{D}_{\text{eval}}$. Following Zhu et al. (2024), we quantify the relative performance degradation caused by perturbations using the **Performance Drop Rate (PDR)**:

$$\text{PDR}(f) \stackrel{def}{=} \frac{\text{Acc}(f, \mathcal{D}_{\text{eval}}) - \text{Acc}(f, \delta(\mathcal{D}_{\text{eval}}))}{\text{Acc}(f, \mathcal{D}_{\text{eval}})},$$

where $\delta(\mathcal{D}_{\text{eval}})$ denotes the evaluation set in which perturbations are applied to the input image before being processed by the model. A higher PDR value indicates greater performance degradation under noise, while a lower value suggests stronger robustness.

## 4 Experiments

### 4.1 Evaluation Models and Evaluation Datasets

We conduct a comprehensive robustness evaluation of the two paradigms, Standard VLM and Visual CoT VLM, under input perturbations. We use two representative VLMs, LLaVA-1.5-7b (Liu et al., 2023a) and VisCoT-7b-224 (Shao et al., 2024), each evaluated under both paradigms by toggling the use of Visual CoT reasoning.

Our evaluation spans four widely adopted datasets across both natural and document-based VQA tasks: CUB (Wah et al., 2011), SROIE (Huang et al., 2019), DocVQA (Mathew et al., 2021), and TextCaps (Sidorov et al., 2020). For each dataset, we construct corresponding perturbed variants

Table 1: PDR (%) of two VLM paradigms (**Standard VLMs** vs. **Visual CoT VLMs**) under 12 image perturbations across four datasets and two base models (LLaVA-1.5-7b, VisCoT-7b-224).

| Dataset | Model | Paradigm | Gaussian | Shot | Impulse | Defocus | Zoom | Pixelate | Elastic | Contrast | BIM | FGSM | PGD | C&W |
|---|---|---|---|---|---|---|---|---|---|---|---|---|---|---|---|
| CUB | LLaVA-1.5-7b | Standard | -10.3 | -3.4 | -3.4 | -10.3 | -13.8 | 3.4 | -10.3 | -10.3 | 10.3 | 10.3 | 6.9 | 13.8 |
| | | VisCoT | 13.2 | 5.3 | 10.5 | 7.9 | 5.3 | -2.6 | 13.2 | 23.7 | 10.5 | 10.5 | 2.6 | 13.1 |
| | VisCoT-7b-224 | Standard | 0.0 | -6.9 | -10.3 | -10.3 | -13.8 | 3.4 | -3.4 | -10.3 | 3.4 | 1.0 | 6.8 | 3.4 |
| | | Visual CoT | 2.6 | 2.6 | 10.5 | 7.9 | 5.3 | -2.6 | 10.5 | 23.7 | 13.1 | 15.7 | 2.6 | 15.8 |
| SROIE | LLaVA-1.5-7b | Standard | 11.1 | 22.2 | -11.1 | 0.0 | 77.8 | 44.4 | -88.9 | 7.3 | 77.8 | 77.8 | 66.7 | 77.8 |
| | | Visual CoT | 27.3 | 18.2 | 12.1 | 9.4 | 90.9 | 9.1 | 9.1 | 34.8 | 68.5 | 75.8 | 72.7 | 73.9 |
| | VisCoT-7b-224 | Standard | 30.0 | 40.0 | 0.0 | 10.0 | 50.0 | 40.0 | -20.0 | 60.0 | 90.0 | 80.0 | 30.0 | 96.0 |
| | | Visual CoT | 25.2 | 16.5 | -0.7 | 19.4 | 28.1 | 10.8 | 13.7 | 65.5 | 82.7 | 65.4 | 30.9 | 94.2 |
| DocVQA | LLaVA-1.5-7b | Standard | 35.3 | 11.8 | -11.8 | 23.5 | 58.8 | -5.9 | 29.4 | 29.4 | 41.2 | 35.2 | 47.0 | 17.6 |
| | | Visual CoT | -25.7 | 4.8 | -7.6 | 23.8 | 38.1 | 9.5 | 50.0 | 4.8 | 41.9 | 23.8 | 33.3 | 40.9 |
| | VisCoT-7b-224 | Standard | 39.1 | 13.0 | -4.3 | 47.8 | 56.5 | 4.3 | 13.0 | 47.8 | 30.4 | 7.8 | 56.5 | 25.2 |
| | | Visual CoT | -10.2 | 9.1 | 1.9 | 42.1 | 56.6 | -9.4 | 29.4 | 26.0 | 54.3 | 52.8 | 47.1 | 73.2 |
| TextCaps | LLaVA-1.5-7b | Standard | 3.8 | 7.7 | 13.5 | 30.8 | 71.9 | 7.7 | 17.3 | 30.8 | 19.2 | 25.0 | 63.4 | 15.4 |
| | | Visual CoT | 18.0 | 15.6 | 22.7 | 49.0 | 64.3 | 14.8 | 32.6 | 25.3 | 32.6 | 39.9 | 69.1 | 28.5 |
| | VisCoT-7b-224 | Standard | 1.8 | 2.5 | -1.8 | -12.5 | -16.1 | 4.3 | -3.6 | 8.9 | 33.9 | 33.9 | 69.6 | 35.7 |
| | | Visual CoT | 25.0 | 13.2 | 20.6 | 10.3 | 14.7 | 7.4 | 17.6 | 20.6 | 39.7 | 44.8 | 70.5 | 44.1 |

Table 2: Answer Accuracy (%) of two VLM paradigms (**Standard VLMs** vs. **Visual CoT VLMs**) under 12 image perturbations across four datasets and two base models (LLaVA-1.5-7b, VisCoT-7b-224).

| Dataset | Model | Paradigm | Clean | Gaussian | Shot | Impulse | Defocus | Zoom | Pixelate | Elastic | Contrast | BIM | FGSM | PGD | C&W |
|---|---|---|---|---|---|---|---|---|---|---|---|---|---|---|---|---|
| CUB | LLaVA-1.5-7b | Standard | 58.0 | 64.0 | 60.0 | 60.0 | 64.0 | 66.0 | 56.0 | 64.0 | 64.0 | 64.0 | 64.0 | 54.0 | 66.0 |
| | | VisCoT | 76.0 | 66.0 | 72.0 | 68.0 | 70.0 | 72.0 | 78.0 | 66.0 | 58.0 | 68.0 | 68.0 | 74.0 | 66.0 |
| | VisCoT-7b-224 | Standard | 58.0 | 58.0 | 62.0 | 64.0 | 64.0 | 66.0 | 56.0 | 60.0 | 64.0 | 56.0 | 58.0 | 54.0 | 56.0 |
| | | Visual CoT | 76.0 | 74.0 | 74.0 | 68.0 | 70.0 | 72.0 | 78.0 | 68.0 | 58.0 | 66.0 | 64.0 | 74.0 | 64.0 |
| SROIE | LLaVA-1.5-7b | Standard | 9.0 | 8.0 | 7.0 | 10.0 | 9.0 | 2.0 | 5.0 | 17.0 | 8.3 | 2.0 | 2.0 | 3.0 | 2.0 |
| | | Visual CoT | 33.0 | 24.0 | 27.0 | 29.0 | 29.9 | 30.0 | 30.0 | 30.0 | 21.5 | 10.4 | 8.0 | 9.0 | 8.6 |
| | VisCoT-7b-224 | Standard | 10.0 | 7.0 | 6.0 | 10.0 | 9.0 | 5.0 | 6.0 | 12.0 | 4.0 | 1.0 | 2.0 | 7.0 | 0.4 |
| | | Visual CoT | 34.8 | 26.0 | 29.0 | 35.0 | 28.0 | 25.0 | 31.0 | 30.0 | 12.0 | 6.0 | 12.0 | 24.0 | 2.0 |
| DocVQA | LLaVA-1.5-7b | Standard | 17.0 | 11.0 | 15.0 | 19.0 | 13.0 | 7.0 | 18.0 | 12.0 | 12.0 | 10.0 | 11.0 | 9.0 | 14.0 |
| | | Visual CoT | 21.0 | 26.4 | 20.0 | 22.6 | 16.0 | 13.0 | 19.0 | 10.5 | 20.0 | 12.2 | 16.0 | 14.0 | 12.4 |
| | VisCoT-7b-224 | Standard | 11.5 | 7.0 | 10.0 | 12.0 | 6.0 | 5.0 | 11.0 | 10.0 | 6.0 | 8.0 | 10.6 | 5.0 | 8.6 |
| | | Visual CoT | 26.5 | 29.2 | 24.1 | 26.0 | 15.3 | 11.5 | 29.0 | 18.7 | 19.6 | 12.1 | 12.5 | 14.0 | 7.1 |
| TextCaps | LLaVA-1.5-7b | Standard | 52.0 | 50.0 | 48.0 | 45.0 | 36.0 | 14.6 | 48.0 | 43.0 | 36.0 | 42.0 | 39.0 | 19.0 | 44.0 |
| | | Visual CoT | 61.6 | 50.5 | 52.0 | 47.6 | 31.4 | 22.0 | 52.5 | 41.5 | 46.0 | 41.5 | 37.0 | 19.0 | 44.0 |
| | VisCoT-7b-224 | Standard | 56.0 | 55.0 | 54.6 | 57.0 | 63.0 | 65.0 | 53.6 | 58.0 | 51.0 | 37.0 | 37.0 | 17.0 | 36.0 |
| | | Visual CoT | 68.0 | 51.0 | 59.0 | 54.0 | 61.0 | 58.0 | 63.0 | 56.0 | 54.0 | 41.0 | 37.5 | 20.0 | 38.0 |

by applying the image perturbation techniques described in Section 3.1, resulting in 48 image perturbation evaluation splits (e.g., CUB-Gaussian_Noise, CUB-Shot_Noise, SROIE-Gaussian_Noise, SROIE-Shot_Noise, etc.).

## 4.2 EVALUATION RESULTS

As summarized in Table 1 and Table 2, our evaluation reveals distinct robustness characteristics between Standard VLMs and their Visual CoT-enhanced counterparts when subjected to image corruptions:

(1) Visual CoT VLMs exhibit a higher PDR than Standard VLMs in 70 out of 96 evaluated settings. Specifically, the average PDR of Visual CoT VLMs reaches 26.3%, while that of Standard VLMs is only 18.6%. This indicates that Visual CoT VLMs are generally more vulnerable to perturbations compared to Standard VLMs. The trend is particularly pronounced on the TextCaps dataset, where Visual CoT VLMs exhibit higher performance degradation than the Standard VLMs in 100% of perturbation cases.

(2) Although Visual CoT VLMs exhibit lower robustness than Standard VLMs in terms of PDR, their accuracy under perturbations remains higher in 79 out of 96 cases. Notably, on the CUB and TextCaps datasets, Standard VLMs occasionally show an apparent accuracy increase under perturbations; however, even in these cases, the resulting accuracy still falls below that of the perturbed Visual CoT VLMs.

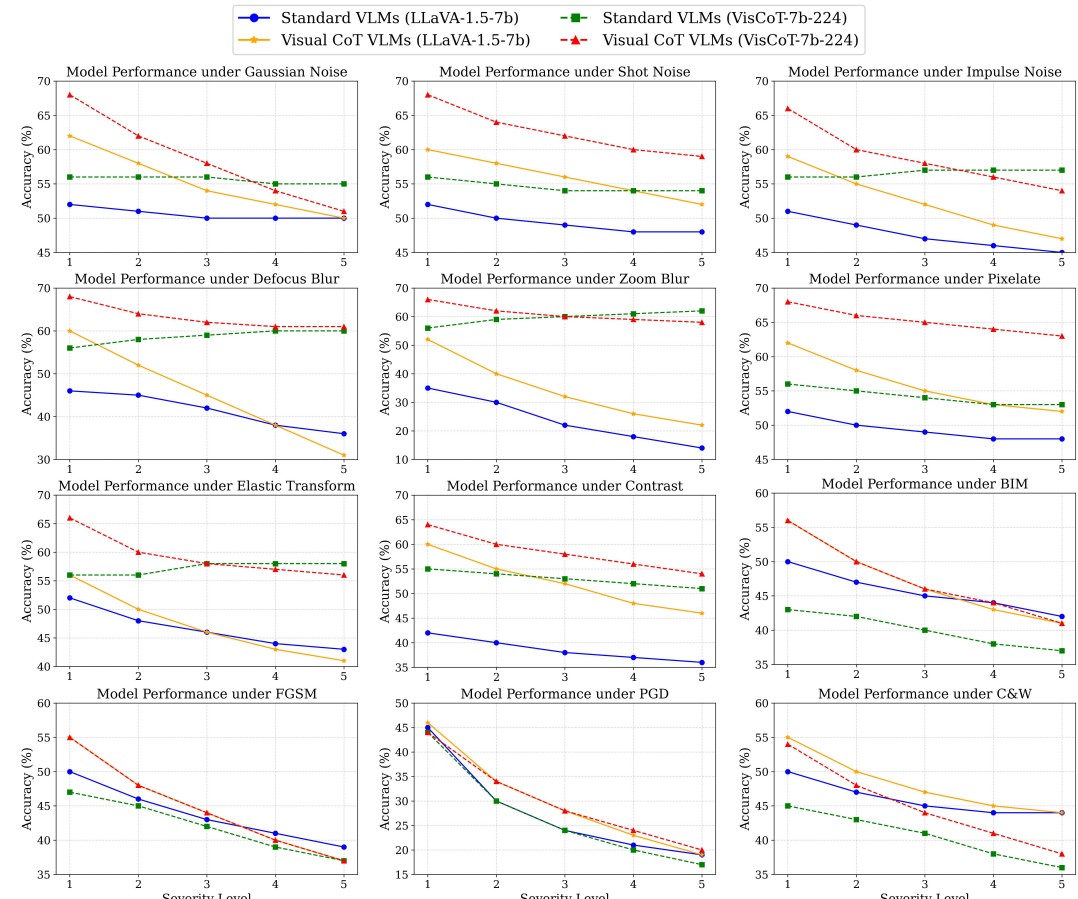

Figure 2: Accuracy trends of Visual CoT VLMs and Standard VLMs under varying image perturbation severity levels.

### 4.3 PERFORMANCE TRENDS ACROSS PERTURBATION SEVERITY LEVELS

To explore how model accuracy evolves under increasing levels of image corruption, we conduct a severity-aware evaluation across 5 levels of perturbation intensity. Figure 2 presents representative accuracy degradation curves across all perturbation types on the CUB dataset using the LLaVA-1.5-7b model.

As illustrated, the Visual CoT VLMs' performance curve exhibits a steeper decline compared to Standard VLMs' as noise severity increases. This indicates that Visual CoT VLMs are more sensitive to perturbation, with performance dropping more rapidly under increasingly severe corruption. However, despite this higher degradation rate, Visual CoT VLMs typically maintain a higher absolute accuracy across all severity levels. This suggests that while they are less robust, their overall capacity for accurate reasoning remains stronger than that of the Standard VLMs. This pattern aligns with our previous quantitative findings on PDR and perturbed accuracy.

## 5 ANALYSIS

### 5.1 WHY ARE VISUAL COT VLMS MORE FRAGILE?

The experimental results reveal a consistent pattern: although Visual CoT VLMs achieve higher absolute accuracy than Standard VLMs baseline, they suffer a more severe accuracy drop under the same perturbation conditions. We attribute this to fundamental differences in their reasoning paradigms.

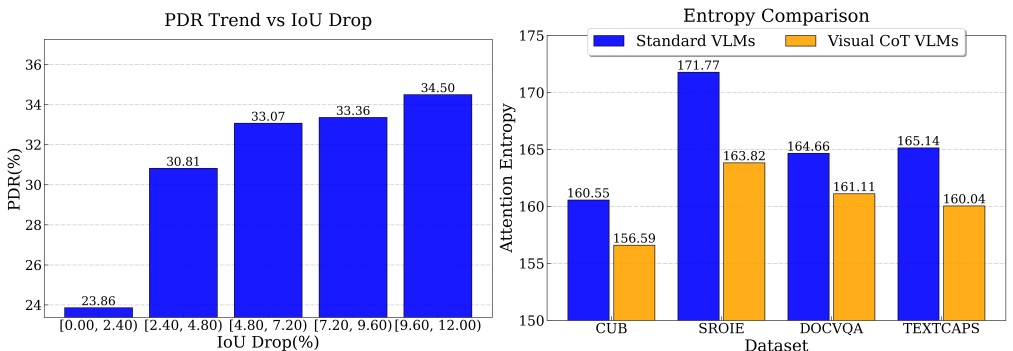

Figure 3: Correlation between PDR and IoU Degradation.

Figure 4: Average Attention Entropy.

Standard VLMs adopt a single-step inference paradigm, directly mapping the input image and question to an answer, which reduces noise propagation and makes them less sensitive to perturbations. In contrast, Visual CoT VLMs employ a multi-stage chain-of-thought framework that integrates global and localized visual information. This enhances reasoning precision but also increases complexity, causing input perturbations to cascade through multiple steps and lead to sharper accuracy degradation.

To investigate, we further analyze the relationship between the accuracy of reasoning steps and the final model performance. In particular, we measure the quality of intermediate bounding box predictions using Intersection over Union (IoU, as described in Appendix A.4), and examine how variations in IoU relate to the PDR of the overall system. The results (as shown in Figure 3) reveal a clear positive correlation: when intermediate localization accuracy decreases, the final prediction accuracy also drops more severely. This indicates that errors accumulated in the intermediate reasoning stages propagate through the Chain-of-Thought process, thereby amplifying the overall fragility of Visual CoT.

## 5.2 CAN ATTENTION MAPS EXPLAIN WHY VISUAL COT VLMS PERFORMS BETTER UNDER PERTURBATION?

To better understand why Visual CoT VLMs achieve higher accuracy than Standard VLMs under noisy image conditions, we analyze the 3D attention distributions obtained from perturbed inputs, as shown in Figure 5. These visualizations reveal a clear distinction between the two paradigms: while Standard VLMs tend to spread attention across broader regions with multiple dispersed peaks, Visual CoT VLMs produce more concentrated and sharper attention peaks focused on specific regions of the input.

This visual difference indicates that Visual CoT VLMs allocate their attention more selectively, focusing on semantically relevant areas while suppressing irrelevant regions. To quantitatively support this observation, we compute the attention entropy for each data instance. Specifically, a lower entropy value indicates that the model's attention is concentrated on more specific regions, whereas a higher entropy suggests a more dispersed focus. As shown in Figure 4, Visual CoT VLMs consistently exhibits lower entropy across samples, confirming a narrower and more focused attention distribution compared to Standard VLMs.

Such concentrated attention behavior helps Visual CoT VLMs better withstand noisy inputs by minimizing distractions from irrelevant tokens. In contrast, the broader and more uniform attention of Standard VLMs may dilute the impact of informative cues, reducing answer reliability under noise.

## 6 ROBUSTNESS ENHANCEMENT

Based on the above analysis, we argue that the multi-step reasoning process in Visual CoT VLMs acts as a double-edged sword: while it drives superior performance through structured reasoning and multimodal integration, it also introduces additional vulnerability to perturbations due to longer

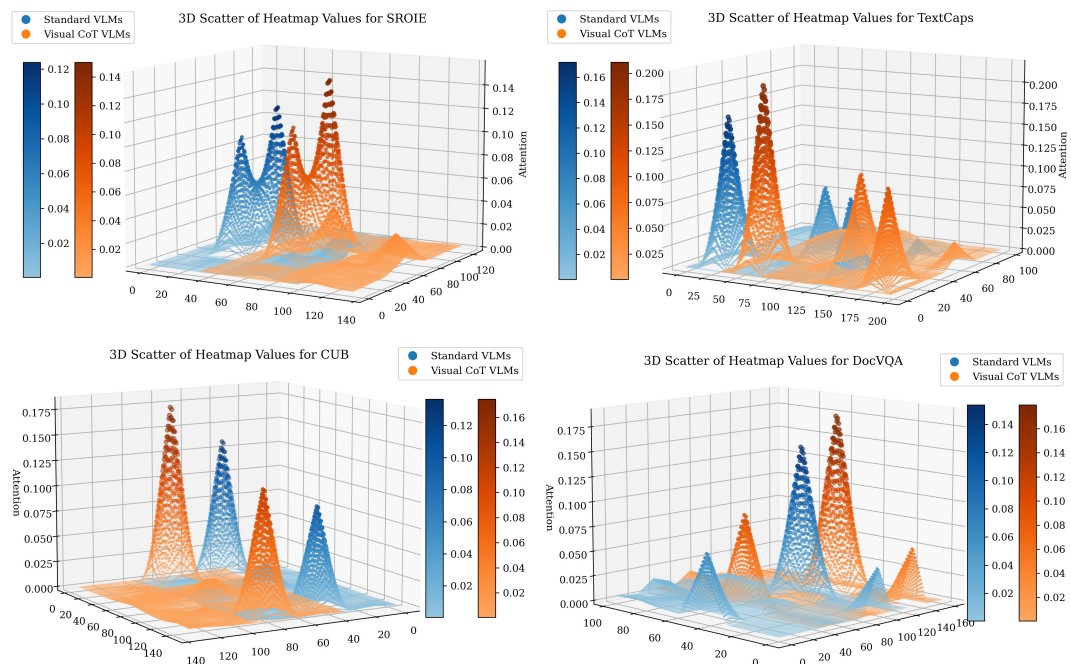

Figure 5: 3D scatter plots of attention scores under noise conditions for Standard and Visual CoT VLMs. Visual CoT exhibits more concentrated attention over key regions.

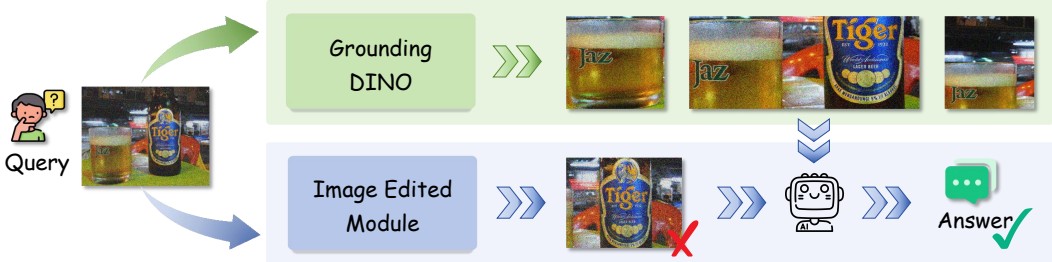

Figure 6: Enhanced Visual CoT pipeline with Grounding DINO.

reasoning chains. Among these, the intermediate components, namely the local image patches play a pivotal role. Consequently, strengthening the components offers a promising and feasible direction to enhance Visual CoT VLMs ' robustness against noisy perturbations. In the following, we present a new approach to achieving this goal.

## 6.1 INCORPORATING GROUNDING DINO FOR ENHANCED VISUAL INFORMATION

To enhance the robustness of Visual CoT VLMs under visual perturbations, we incorporate an auxiliary visual grounding step into the reasoning pipeline in a plug-and-play manner (as shown in Figure 6). This step complements the original single-region strategy by identifying multiple semantically relevant regions that may provide redundant visual cues under noisy conditions.

Specifically, given an image–question pair, we apply Grounding DINO to generate text-conditioned region proposals. All bounding boxes with confidence scores exceeding a threshold (typically 0.4) are retained, and the selected regions are cropped from the original image. Then these auxiliary patches are appended to the original visual inputs and subsequently used within Visual CoT VLMs as supplementary visual cues to assist answer generation. This design encourages VLMs to attend to diverse visual perspectives during multi-step reasoning, thereby improving robustness under perturbations.

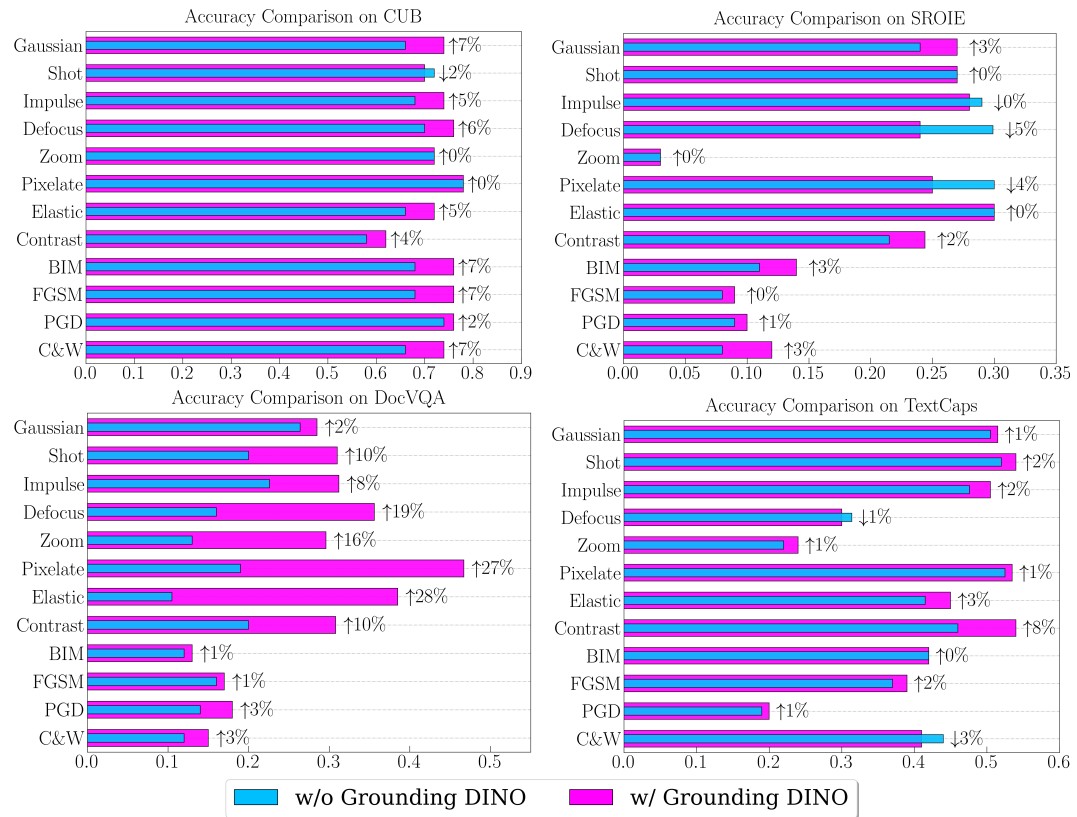

Figure 7: Accuracy comparison of Visual CoT with and without Grounding DINO under different image perturbations across four datasets.

## 6.2 EXPERIMENTAL RESULTS AND ANALYSIS

Experimental results in Figure 7 demonstrate that, across all types of visual perturbations, the integration of Grounding DINO consistently enhances Visual CoT VLMs performance, yielding noticeable accuracy gains on the majority of evaluated datasets. On average, the incorporation of Grounding DINO leads to a 6% increase in accuracy. The improvements are most pronounced on DocVQA, where accuracy gains frequently exceed 10% under perturbations such as Pixelate, Elastic Transformations and Contrast Adjustments.

This improvement can be primarily attributed to Grounding DINO's ability to identify key regions in images. By integrating target bounding box information relevant to the given question into the Visual CoT reasoning process, the system's ability to resist noise interference and extract critical information is effectively strengthened. These findings suggest that incorporating visual grounding model into the reasoning process can significantly enhance the robustness of Visual CoT VLMs under perturbations.

## 7 CONCLUSION

In this paper, we present a systematic robustness study of Visual CoT reasoning in VLMs, revealing a fundamental trade-off: while Visual CoT improves answer accuracy, it also introduces increased sensitivity to visual perturbations. To address this limitation, we introduce a plug-and-play enhancement based on the Grounding DINO model, which improves robustness without requiring retraining of the base VLMs. Our work provides a foundation for future research on developing robust and reliable multimodal reasoning systems.

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

# A    APPENDIX

## A.1    USE OF LARGE LANGUAGE MODELS

To improve the readability of this manuscript, we used LLMs for language polishing, such as rephrasing sentences for clarity and correcting grammar. The LLMs were not involved in designing the methodology, conducting experiments, analyzing results, or drawing scientific conclusions. All substantive research contributions are the sole work of the authors.

## A.2    SEVERITY LEVELS OF NATURAL PERTURBATIONS

For all natural corruption methods, we adopt five severity levels from 1 to 5, following the ImageNet-C benchmark design. Severity controls the intensity of distortion, with level 1 corresponding to the weakest corruption and level 5 to the strongest. Below we explain the parameterization of each corruption and how these parameters influence the degree of degradation.

Table 3: Severity Level Settings for Visual Perturbations (Code-based Implementation). Severity increases from 1 to 5, with higher levels indicating stronger perturbations.

| Perturbation | Parameter | Severity Values (1→5) | Effect Description |
|---|---|---|---|
| Gaussian Noise | Std. dev. $\sigma$ | [0.08, 0.12, 0.18, 0.26, 0.38] | Adds Gaussian-distributed pixel noise. Larger $\sigma \rightarrow$ stronger random fluctuations, fine details vanish at severity 5. |
| Shot Noise | Photon count scale $c$ | [60, 25, 12, 5, 3] | Lower $c \rightarrow$ stronger Poisson noise. Severity 5 simulates extreme low-light, heavy discrete fluctuations. |
| Impulse Noise | Pixel corruption ratio $p$ | [0.03, 0.06, 0.09, 0.17, 0.27] | Higher $p$ replaces more pixels with black/white. Severity 5 $\rightarrow$ ~27% pixels corrupted. |
| Defocus Blur | Disk radius, alias blur | [(3,0.1), (4,0.5), (6,0.5), (8,0.5), (10,0.5)] | Increasing radius $\rightarrow$ stronger out-of-focus blur. At severity 5, edges/boundaries disappear. |
| Zoom Blur | Zoom factor ranges $c$ | 1: 1.00–1.10 (step 0.01)
2: 1.00–1.15 (step 0.01)
3: 1.00–1.21 (step 0.02)
4: 1.00–1.26 (step 0.02)
5: 1.00–1.33 (step 0.03) | Combines zoomed-in frames. Larger ranges represent stronger radial streaks. Severity 5 represents heavy smearing. |
| Pixelate | Downsample ratio $c$ | [0.6, 0.5, 0.4, 0.3, 0.25] | Image is resized to $c\cdot$ original then upscaled. Lowe r values $\rightarrow$ larger blocks. Severity 5 $\rightarrow$ coarse blockiness. |
| Elastic Transform | $(\alpha, \sigma, \alpha_{affine})$ | 1: (224×2, 224×0.7, 224×0.1)
2: (224×2, 224×0.08, 224×0.2)
3: (224×0.05, 224×0.01, 224×0.02)
4: (224×0.07, 224×0.01, 224×0.02)
5: (224×0.12, 224×0.01, 224×0.02) | $\alpha$ controls displacement, $\sigma$ smooths deformation, $\alpha_{affine}$ adds affine distortion. Higher severity $\rightarrow$ stronger warping, shapes deformed. |
| Contrast Reduction | Contrast scaling $c$ | [0.4, 0.3, 0.2, 0.1, 0.05] | Lower $c$ reduces pixel variance. Severity 5 $\rightarrow$ image looks flat/washed out. |

## A.3    SEVERITY LEVELS OF WHITE-BOX ADVERSARIAL ATTACK

The four standard adversarial attacks: FGSM, BIM, PGD, and C&W, are conducted on the image encoder of the model (e.g., CLIP ViT or other vision towers), targeting the embedding similarity between clean and adversarial samples. The goal is to cause minimal pixel changes while maximally deviating the internal representations Gao et al. (2024).

### A.3.1 FAST GRADIENT SIGN METHOD (FGSM)

---

**Algorithm 1** FGSM Algorithm

---

**Require:** Input image $x$, vision encoder $f(\cdot)$, loss function $\mathcal{L}$ (e.g., MSE), perturbation bound $\epsilon$
**Ensure:** Adversarial image $x_{\text{adv}}$
 1: Compute clean embedding: $z_{\text{clean}} \leftarrow f(x)$
 2: Initialize perturbation: $\delta \leftarrow 0$, set $\delta$ as a trainable tensor
 3: $z_{\text{adv}} \leftarrow f(x + \delta)$
 4: $\mathcal{L}_{\text{adv}} \leftarrow \text{MSE}(z_{\text{adv}}, z_{\text{clean}})$
 5: Compute gradient: $\nabla_\delta \mathcal{L}_{\text{adv}}$
 6: Update perturbation: $\delta \leftarrow \epsilon \cdot \text{sign}(\nabla_\delta \mathcal{L}_{\text{adv}})$
 7: $x_{\text{adv}} \leftarrow \text{clip}(x + \delta, 0, 1)$ ▷ Ensure valid pixel range
 8: **return** $x_{\text{adv}}$

---

Table 4: Severity Level Settings for FGSM Attack

| Severity Level | $\epsilon$ (Perturbation Magnitude) |
|:---:|:---:|
| 1 | $\frac{1}{255}$ |
| 2 | $\frac{2}{255}$ |
| 3 | $\frac{4}{255}$ |
| 4 | $\frac{6}{255}$ |
| 5 | $\frac{8}{255}$ |

### A.3.2 BASIC ITERATIVE METHOD (BIM)

---

**Algorithm 2** BIM Algorithm

---

**Require:** Input image $x$, vision encoder $f(\cdot)$, loss function $\mathcal{L}$ (e.g., MSE), step size $\alpha$, maximum perturbation $\epsilon$, number of iterations $T$
**Ensure:** Adversarial image $x_{\text{adv}}$
 1: Compute clean embedding $z_{\text{clean}} \leftarrow f(x)$
 2: Initialize perturbation $\delta \leftarrow \mathbf{0}$ ▷ No random start
 3: **for** $t = 1$ to $T$ **do**
 4:     $z_{\text{adv}} \leftarrow f(x + \delta)$ ▷ Get perturbed embedding
 5:     $\mathcal{L}_{\text{adv}} \leftarrow \text{MSE}(z_{\text{adv}}, z_{\text{clean}})$
 6:     Compute gradient $\nabla_\delta \mathcal{L}_{\text{adv}}$
 7:     $\delta \leftarrow \delta + \alpha \cdot \text{sign}(\nabla_\delta \mathcal{L}_{\text{adv}})$
 8:     $\delta \leftarrow \text{clip}(\delta, -\epsilon, \epsilon)$ ▷ Clip to $\ell_\infty$ ball
 9:     $\delta \leftarrow \text{clip}(x + \delta, 0, 1) - x$ ▷ Ensure pixel validity
10: **end for**
11: $x_{\text{adv}} \leftarrow \text{clip}(x + \delta, 0, 1)$
12: **return** $x_{\text{adv}}$

---

Table 5: Severity Level Settings for BIM Attack

| Severity Level | $\epsilon$ | $\alpha$ (Step Size) | $T$ (Iterations) |
|:---:|:---:|:---:|:---:|
| 1 | $\frac{1}{255}$ | $\frac{0.2}{255}$ | 100 |
| 2 | $\frac{2}{255}$ | $\frac{0.4}{255}$ | 200 |
| 3 | $\frac{4}{255}$ | $\frac{0.8}{255}$ | 300 |
| 4 | $\frac{6}{255}$ | $\frac{1.0}{255}$ | 400 |
| 5 | $\frac{8}{255}$ | $\frac{1.2}{255}$ | 500 |

### A.3.3 PROJECTED GRADIENT DESCENT (PGD)

---
**Algorithm 3** PGD Algorithm

---
**Require:** Input image $x$, vision encoder $f(\cdot)$, loss function $\mathcal{L}$ (e.g., MSE), step size $\alpha$, maximum perturbation $\epsilon$, number of iterations $T$
**Ensure:** Adversarial image $x_{\text{adv}}$
 1: Initialize perturbation $\delta \sim \text{Uniform}(-\epsilon, \epsilon)$
 2: **for** $t = 1$ to $T$ **do**
 3:     $z_{\text{adv}} \leftarrow f(x + \delta)$                          ▷ Get perturbed embedding
 4:     $z_{\text{clean}} \leftarrow f(x)$                    ▷ (Optional) Use precomputed clean embedding
 5:     $\mathcal{L}_{\text{adv}} \leftarrow \text{MSE}(z_{\text{adv}}, z_{\text{clean}})$
 6:     Compute gradient $\nabla_\delta \mathcal{L}_{\text{adv}}$
 7:     $\delta \leftarrow \delta + \alpha \cdot \text{sign}(\nabla_\delta \mathcal{L}_{\text{adv}})$
 8:     $\delta \leftarrow \text{clip}(\delta, -\epsilon, \epsilon)$                          ▷ Project onto $\ell_\infty$ ball
 9: **end for**
10: $x_{\text{adv}} \leftarrow \text{clip}(x + \delta, 0, 1)$                          ▷ Clamp to valid pixel range
11: **return** $x_{\text{adv}}$

---

Table 6: Severity Level Settings for PGD Attack

| **Severity Level** | $\epsilon$ | $\alpha$ (Step Size) | $T$ (Iterations) |
|:---:|:---:|:---:|:---:|
| 1 | $\frac{1}{255}$ | $\frac{0.2}{255}$ | 100 |
| 2 | $\frac{2}{255}$ | $\frac{0.4}{255}$ | 200 |
| 3 | $\frac{4}{255}$ | $\frac{0.8}{255}$ | 300 |
| 4 | $\frac{6}{255}$ | $\frac{1.0}{255}$ | 400 |
| 5 | $\frac{8}{255}$ | $\frac{1.2}{255}$ | 500 |

### A.3.4 CARLINI & WAGNER (C&W) ATTACK (UNTARGETED)

---
**Algorithm 4** C&W Attack Algorithm

---
**Require:** Input image $x$, vision encoder $f(\cdot)$, loss function $\mathcal{L}$ (e.g., MSE), regularization coefficient $C$, learning rate $\eta$, number of iterations $T$
**Ensure:** Adversarial image $x_{\text{adv}}$
 1: Compute clean embedding: $z_{\text{clean}} \leftarrow f(x)$
 2: Convert $x$ to tanh-space: $w \leftarrow \text{arctanh}(2x - 1)$                          ▷ Ensure differentiability
 3: **for** $t = 1$ to $T$ **do**
 4:     $x_{\text{adv}} \leftarrow 0.5 \cdot (\tanh(w) + 1)$                          ▷ Map $w$ back to $[0, 1]$
 5:     $z_{\text{adv}} \leftarrow f(x_{\text{adv}})$
 6:     Compute adversarial loss: $\mathcal{L}_{\text{embed}} \leftarrow \text{MSE}(z_{\text{adv}}, z_{\text{clean}})$
 7:     Compute distortion loss: $\mathcal{L}_{\text{l2}} \leftarrow \|x_{\text{adv}} - x\|_2^2$
 8:     Total loss: $\mathcal{L}_{\text{total}} \leftarrow C \cdot \mathcal{L}_{\text{embed}} + \mathcal{L}_{\text{l2}}$
 9:     Update $w$ via Adam: $w \leftarrow w - \eta \cdot \nabla_w \mathcal{L}_{\text{total}}$
10: **end for**
11: **return** $x_{\text{adv}}$

---

Table 7: Severity Level Settings for C&W Attack

| Severity Level | $C$ (Embed Weight) | $\eta$ (Learning Rate) | $T$ (Iterations) |
|:---:|:---:|:---:|:---:|
| 1 | 0.1 | $1 \times 10^{-3}$ | 100 |
| 2 | 0.5 | $1 \times 10^{-3}$ | 200 |
| 3 | 1.0 | $5 \times 10^{-4}$ | 300 |
| 4 | 2.0 | $1 \times 10^{-4}$ | 400 |
| 5 | 5.0 | $1 \times 10^{-4}$ | 500 |

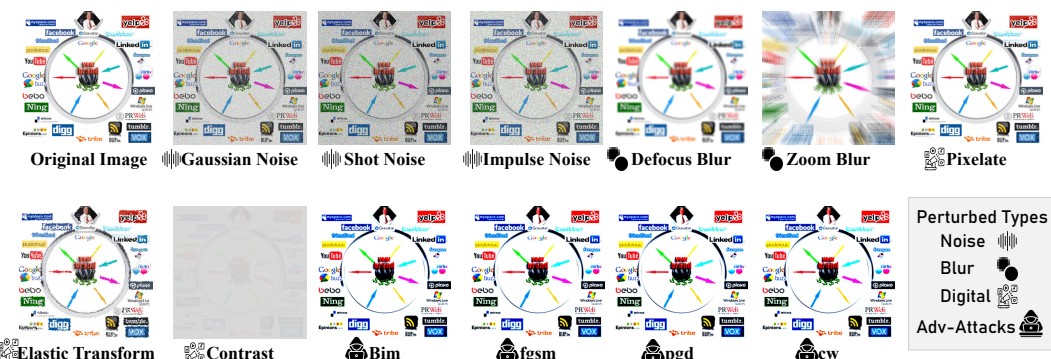

Figure 8: Example images under 8 natural image perturbations and 4 white-box adversarial attacks. The original image is taken from the Flickr30k dataset and shown on the top left.

## A.4 INTERSECTION OVER UNION

In our experiments, we evaluate the accuracy of intermediate localization using the Intersection over Union (IoU) metric. Given two axis-aligned boxes $B_1 = [x_1^{(1)}, y_1^{(1)}, x_2^{(1)}, y_2^{(1)}]$ (prediction) and $B_2 = [x_1^{(2)}, y_1^{(2)}, x_2^{(2)}, y_2^{(2)}]$ (ground truth), IoU is defined as

$$\text{IoU}(B_1, B_2) = \frac{|B_1 \cap B_2|}{|B_1 \cup B_2|}.$$

where $|B_1 \cap B_2|$ denotes the area of overlap between the two boxes, and $|B_1 \cup B_2|$ represents their union area. The overlap region is determined by taking the maximum of the top-left coordinates and the minimum of the bottom-right coordinates. The area of each box is then computed as the product of its width and height, and the union is given by the sum of both areas minus the intersection.

This implementation ensures that IoU ranges between 0 and 1, where 0 indicates no overlap and 1 denotes perfect alignment. In practice, a higher IoU signifies more accurate localization of predicted regions with respect to ground-truth annotations, while lower values reflect misalignment or degraded localization quality.

## A.5 INTERMEDIATE PERTURBATION REUSLTS

We conduct a supplementary experiment that perturbs not only the input image but also the intermediate local patches. This leads to a more pronounced performance drop, underscoring the sensitivity of Visual CoT VLMs to noise in intermediate components.

Table 8: Answer accuracy (%) of Visual CoT models under different perturbation positions: either applied on the global image only ("Global Only") or both on the intermediate local crops ("Global and Local"). Experiments are conducted under severity level 5 across four datasets.

| Dataset | Model | Perturb Location | Gaussian | Shot | Impulse | Defocus | Zoom | Pixelate | Elastic | Contrast | BIM | FGSM | PGD | C&W |
|---|---|---|---|---|---|---|---|---|---|---|---|---|---|---|
| **CUB** | LLaVA-1.5-7b | Global Only | 66.0 | 72.0 | 68.0 | 70.0 | 72.0 | 78.0 | 66.0 | 58.0 | 68.0 | 68.0 | 74.0 | 66.0 |
| | | Global and Local | **60.5** | **65.3** | **61.7** | **64.0** | **66.2** | **70.8** | **59.0** | **50.2** | **62.2** | **67.0** | **62.1** | **58.2** |
| | VisCoT-7b-224 | Global Only | 74.0 | 74.0 | 68.0 | 70.0 | 72.0 | 78.0 | 68.0 | 58.0 | 50.0 | 54.0 | 46.0 | 40.0 |
| | | Global and Local | **66.3** | **67.8** | **60.5** | **62.6** | **67.1** | **69.4** | **60.3** | **48.7** | **42.1** | **47.0** | **39.0** | **35.0** |
| **SROIE** | LLaVA-1.5-7b | Global Only | 24.0 | 27.0 | 29.0 | 29.9 | 30.0 | 30.0 | 30.0 | 21.5 | 10.4 | 8.0 | **9.0** | **8.6** |
| | | Global and Local | **20.3** | **23.5** | **25.0** | **26.2** | **25.8** | **27.2** | **25.1** | **17.0** | **14.6** | **16.2** | 12.5 | 9.3 |
| | VisCoT-7b-224 | Global Only | 26.0 | 29.0 | 35.0 | 28.0 | 25.0 | 31.0 | 30.0 | 12.0 | **6.0** | **12.0** | 24.0 | **2.0** |
| | | Global and Local | **20.8** | **25.3** | **29.2** | **24.1** | **22.5** | **26.6** | **23.0** | **9.5** | 16.5 | 18.3 | **14.4** | 10.7 |
| **DocVQA** | LLaVA-1.5-7b | Global Only | 26.4 | 20.0 | 22.6 | 16.0 | 13.0 | 19.0 | 10.5 | 20.0 | 12.2 | 16.0 | 14.0 | 12.4 |
| | | Global and Local | **21.2** | **16.7** | **18.0** | **13.4** | **10.6** | **15.8** | **8.1** | **15.5** | **11.6** | **13.0** | **10.0** | **7.2** |
| | VisCoT-7b-224 | Global Only | 29.2 | 24.1 | 26.0 | 15.3 | 11.5 | 29.0 | 18.7 | 19.6 | **12.1** | **12.5** | 14.0 | **7.1** |
| | | Global and Local | **23.7** | **20.2** | **21.8** | **12.6** | **9.3** | **24.5** | **14.2** | **16.0** | 13.2 | 15.6 | **12.2** | 8.8 |
| **TextCaps** | LLaVA-1.5-7b | Global Only | 50.5 | 52.0 | 47.6 | 31.4 | 22.0 | 52.5 | 41.5 | 46.0 | 41.5 | **37.0** | **19.0** | 44.0 |
| | | Global and Local | **44.3** | **47.1** | **42.0** | **27.0** | **18.2** | **46.8** | **35.0** | **38.4** | **34.1** | 37.8 | 30.1 | **25.3** |
| | VisCoT-7b-224 | Global Only | 51.0 | 59.0 | 54.0 | 61.0 | 58.0 | 63.0 | 56.0 | 54.0 | 41.0 | **37.5** | **20.0** | 38.0 |
| | | Global and Local | **44.6** | **52.8** | **47.2** | **54.1** | **49.5** | **56.6** | **49.2** | **46.8** | **39.3** | 42.5 | 35.3 | **30.6** |

