# OpenReview forum: "Visual CoT Makes VLMs Smarter but More Fragile"
_ICLR.cc/2026/Conference — ICLR 2026 Conference Withdrawn Submission_

### Official Review · Reviewer_a72s · 2025-10-25

**Soundness:** 3
**Presentation:** 2
**Contribution:** 3
**Rating:** 4
**Confidence:** 5

**Summary:**

This paper investigates the robustness of Visual CoT Vision-Language Models under various visual perturbations. It presents insightful analyses explaining their higher sensitivity compared to standard VLMs and proposes a practical mitigation strategy that leverages image patch augmentation via DINO to reduce the propagation effects of perturbations.

**Strengths:**

- The paper addresses an important and previously unexplored question: evaluating the robustness of Visual CoT VLMs under perturbations.

- It empirically shows that Visual CoT VLMs are less robust to corruption, with performance degrading more sharply as perturbation severity increases.

- The paper provides a **thorough and well-structured analysis**, including failure case studies, explanations for observed weaknesses, and a proposed practical solution.

- The hypotheses are well justified: for instance, the authors show that perturbations propagate through decreased IoU, which correlates with drops in DPR. Visual CoT models exhibit lower entropy (more focused attention), leading to higher sensitivity when visual regions are corrupted. This careful reasoning strengthens the paper’s analytical depth.

**Weaknesses:**

- **DINO Grounding section**: the experimental setup is not fully clear. Are the DINO-generated patches also subject to perturbations? If not, the authors should report the additional computational overhead this approach introduces, as well as the number of patches used. An ablation study examining how the number of DINO patches affects robustness would strengthen the claims.

- While the DINO-based approach is interesting, it currently appears ad hoc and not scalable. Demonstrating that it is computationally cheap or easy to implement would improve the practical value of the contribution.

- It would also be valuable to propose alternatives that modify the Visual CoT mechanism directly, e.g., adjusting attention entropy by for instance, injecting noise into attention maps to **improve robustness in a more principled way**.

- **Evaluation via GPT-4o**: the paper lacks sufficient details about the evaluation process. How was the LLM-as-a-judge setup validated? What prompts were used? More transparency and justification are needed for the reliability of this evaluation method.

- **Reproducibility**: no code repository is provided, which limits the reproducibility of results and the utility of this work for further research.

- The paper would benefit from **clearer motivation and impact discussion**, e.g., the importance of robustness under real-world noise, low-quality images, or adversarial attacks. Including this context and releasing code would greatly increase the paper’s impact and credibility.


While the paper explores an important and novel research direction and provides valuable insights into Visual CoT sensitivity, the lack of methodological clarity and reproducibility (particularly regarding the GPT-4o evaluation and DINO grounding experiments) weakens its contribution.
Therefore, I recommend rejection in its current form. However, **I would be open to revising my score upward** if the rebuttal provides clearer experimental details and improved validation of the proposed methods.

**Questions:**

Q1: Why does performance degrade even more on the TextCaps dataset? Why do Visual CoT VLMs show higher degradation than standard VLMs across all perturbation types?

Q2: In Figure 5, what explains the spatial shift between the attention peaks of Visual CoT and standard VLMs?

Q3: In Figure 7, why does DINO grounding sometimes worsen robustness (e.g., accuracy drops on SROIE for Defocus/Pixelate and on TextCaps for C&W)? What is the authors’ intuition, and how could it be experimentally validated?

---

### Official Review · Reviewer_BBLK · 2025-10-27

**Soundness:** 3
**Presentation:** 3
**Contribution:** 2
**Rating:** 4
**Confidence:** 4

**Summary:**

## SUMMARY
This paper is the first systematic evaluation of Visual Chain-of-Thought (Visual CoT) robustness in Vision-Language Models (VLMs). It identifies a key trade-off: Visual CoT improves accuracy but increases sensitivity to visual perturbations. The authors attribute this fragility to error propagation from intermediate image patches and explain the higher accuracy via more focused attention. To mitigate the fragility, they propose a plug-and-play solution using Grounding DINO to add redundant visual cues.

**Strengths:**

## Strengths
As the first systematic evaluation of Visual CoT robustness, this work fills a critical research gap. Its strengths include a comprehensive experimental design, an insightful causal analysis, and a practical solution using Grounding DINO.

**Weaknesses:**

## Weaknesses
- The study focuses on a single-crop implementation of Visual CoT. Its conclusions on fragility may not generalize to other Visual CoT paradigms, such as those involving sketch generation or multiple region annotations, thus limiting the findings' scope.
- The experiments are limited to 7B-scale models. The "smarter but more fragile" trade-off may not hold for much larger VLMs (e.g., GPT-4V), which could have greater intrinsic robustness.
- The proposed Grounding DINO solution, while effective, lacks methodological novelty. It is a practical engineering fix—combining off-the-shelf models—rather than a new fundamental technique. The observed gains likely come from the increased fault tolerance of using multiple candidate regions.

**Questions:**

## Questions
- How sensitive is the Grounding DINO enhancement to the confidence threshold of 0.4? What is the fallback strategy if it fails to find relevant regions, and does this create a new failure mode?
- In the adversarial attack setup, perturbations are only applied to the initial input image to ensure a fair comparison. The supplementary experiment in the appendix also shows that if the intermediate generated patches are also perturbed, performance degrades further.  I'm curious how performance degrades if only the intermediate patches are attacked (preserving the global image).
- Could you provide a deeper analysis for the counter-intuitive performance  improvement (negative PDR) seen in Table 1? Is this statistical noise, or a form of regularization from the perturbations?

---

### Official Review · Reviewer_YAZA · 2025-11-01

**Soundness:** 2
**Presentation:** 3
**Contribution:** 2
**Rating:** 4
**Confidence:** 4

**Summary:**

This paper presents the first systematic robustness evaluation of Visual CoT in VLMs. The authors' core contribution is identifying a key trade-off: integrating intermediate visual edits (like cropping) improves absolute accuracy on both clean and corrupted images, but simultaneously makes the models significantly more fragile, leading to sharper performance drops under visual perturbations. The paper traces this fragility to the intermediate reasoning components and proposes a solution using Grounding DINO to enhance robustness by providing redundant visual cues.

**Strengths:**

- Originality: the authors identified and systematically benchmarked a "smarter but more fragile" trade-off in the Visual CoT paradigm.
- The work is of high quality, supported by a rigorous experimental setup with 12 perturbation types across 4 datasets. It links the observed fragility to the degradation of intermediate reasoning components.
- Clarity: The paper is clear, and its findings are significant.
- Significance: It offers a timely insight in VLM reasoning while also providing an immediate, practical solution.

**Weaknesses:**

- The paper's central claim, "Visual CoT Makes VLMs... More Fragile", is a strong generalization. However, the study's implementation of "Visual CoT" is limited to a single "crop-and-fuse" paradigm represented by VisCoT. The paper's own related work section mentions other Visual CoT approaches, such as sketching (Sketchpad) and generative editing (ReFous). It is unclear if the fragility observed is specific to this "crop-and-fuse" method or a fundamental property of all Visual CoT paradigms.
- The experiments are limited to two 7B-scale models (LLaVA-1.5 and VisCoT), which are relatively small by today's standards. These findings may not generalize to larger, more capable models that might have different internal robustness characteristics
- The primary solution involves integrating Grounding DINO (GD) to provide "high-confidence local visual cues" as a robustness enhancement. However, GD is itself a model that is likely susceptible to the visual perturbations. The paper assumes GD will function robustly on corrupted images, but this critical assumption is never tested. If GD's own performance degrades under noise, it may fail to provide stabilizing cues, potentially making the system even more fragile. It would be good to evaluate the performance (e.g., IoU) of GD itself on the perturbed datasets.

**Questions:**

See weaknesses.

---

### Official Review · Reviewer_gyob · 2025-11-02

**Soundness:** 3
**Presentation:** 3
**Contribution:** 3
**Rating:** 4
**Confidence:** 3

**Summary:**

This paper provides the study of Visual Chain-of-Thought in VLMs, testing 12 natural corruptions and 4 adversarial attacks across four VQA datasets. It finds a clear trade-off: Visual CoT yields higher absolute accuracy but larger degradation under noise (average PDR 26.3% vs 18.6% for standard VLMs), with fragility traced to the edited image patches used in intermediate reasoning. A plug-and-play fix—injecting Grounding DINO proposals into the Visual CoT pipeline—improves robustness without retraining (≈+6% accuracy on average under perturbations).

**Strengths:**

I like the study of adversarial attacks in CoT VLMs, and the findings is new to me.

Attention analysis shows Visual CoT concentrates focus on salient regions (explaining higher clean/perturbed accuracy) yet amplifies error propagation when inputs are corrupted, along with other diagnostics such as PDR and IoU–PDR correlation.

Studying on 4 different datasets as well as many natural corruptions and adversarial attacks provide many empirical evidences.

**Weaknesses:**

Based on the Figure 2, many attacks with gradient-based approach shows similar trends for both standard VLMs and CoT VLMs. Also, it is surprised for me to see the CoT VLMs performance drops that much under very simple attacks such as Gaussian noise and blur (if the experiments conducted correctly).

Extending the experiments to newer models, such as qwen, and extending to larger models, such as 72B models would help make the argument more solid. LLaVA-v1.5 is actually a very old model, studying on that, I am unsure if the founding can be generalized to other models.

Overall, this paper does not propose any technical novelty but that is not necessary if the empirical analysis is extensive and systematic.

**Questions:**

Plz address all the concerns raised above.

My biggest concerns is the current experimental results limited on 7B models and on LLaVA-v1.5 and VisCoT. I will consider new experimental results and if other reviewers believe the existing empirical results are good enough.

---

### Note · Authors · 2025-11-26

I have read and agree with the venue's withdrawal policy on behalf of myself and my co-authors.